# Single-Use and Reusable Flexible Bronchoscopes in Pulmonary and Critical Care Medicine

**DOI:** 10.3390/diagnostics12010174

**Published:** 2022-01-12

**Authors:** Elliot Ho, Ajay Wagh, Kyle Hogarth, Septimiu Murgu

**Affiliations:** 1Section of Pulmonary and Critical Care Medicine/Interventional Pulmonology, Department of Medicine, Loma Linda University, Loma Linda, CA 92354, USA; 2Section of Pulmonary and Critical Care Medicine/Interventional Pulmonology, Department of Medicine, The University of Chicago, Chicago, IL 60637, USA; awagh@medicine.bsd.uchicago.edu (A.W.); khogarth@medicine.bsd.uchicago.edu (K.H.); smurgu@medicine.bsd.uchicago.edu (S.M.)

**Keywords:** bronchoscopy, single-use flexible bronchoscope, disposable flexible bronchoscope, reusable flexible bronchoscope, pulmonary, critical care

## Abstract

Flexible bronchoscopy plays a critical role in both diagnostic and therapeutic management of a variety of pulmonary disorders in the bronchoscopy suite and the intensive care unit. In the set-ting of the ongoing viral pandemic, single-use flexible bronchoscopes (SUFB) have garnered attention as various professional pulmonary societies have released guidelines regarding uses for SUFB given the concern for risk of viral transmission when using reusable flexible bronchoscopes (RFB). In addition to offering sterility, SUFBs are portable, easily accessible, and may be more cost-effective than RFB when considering the potential costs of treating bronchoscopy-related infections. Furthermore, since SUFBs are one time use, they do not require reprocessing after use, and therefore may translate to reduced cleaning and storage costs. Despite these advantages, RFBs are still routinely used to perform advanced diagnostic and therapeutic bronchoscopic procedures given the need for optimal maneuverability, handling, angle of deflection, image quality, and larger channel size for passing of ancillary instruments. Here, we review the published evidence on the applications of single-use and reusable bronchoscopes in bronchoscopy suites and intensive care units. Specifically, we will discuss the advantages and disadvantages of these devices as pertinent to fundamental, advanced, and therapeutic bronchoscopic interventions.

## 1. Introduction

Flexible bronchoscopy plays a critical role in diagnostic and therapeutic interventions for a variety of pulmonary disorders. Diagnostic uses include airway inspection and sampling of endobronchial lesions, bronchoalveolar lavage (BAL), brushings, and lung biopsies. Flexible bronchoscopy is commonly used for lung cancer diagnosis and mediastinal staging utilizing endobronchial ultrasound. Therapeutic uses include aspiration of mucoid or hemorrhagic secretions, endobronchial valve placement (for persistent air leak and bronchoscopic lung volume reduction), thermal ablative therapy, cryotherapy, tumor debulking, foreign body retrieval, airway stent deployment, guidance for percutaneous tracheostomy placement, and fiducial marker placement prior to lung resection.

In the setting of the ongoing COVID-19 pandemic, single-use flexible bronchoscopes (SUFB) have garnered attention as various professional pulmonary societies have released guidelines regarding uses for SUFB given the concern for risk of viral transmission when using reusable flexible bronchoscopes (RFB). Early in the pandemic, Pulmonary societies such as Chinese Medical Association (CMA), American Association for Bronchology and Interventional Pulmonology (AABIP), Spanish Society of Pneumology and Thoracic Surgery (SEPAR), and Argentinean Association for Bronchology (AABE) have recommended using SUFB in patients with known or suspected SARS-CoV-2 infection, if these scopes were available [1,2]. However, there are no specific recommendations from these or other societies regarding the optimal type of equipment for patients without known or suspected SARS-CoV-2 infection during the pandemic [3]. Since their initial introduction, there have been improvements in the maneuverability, image quality, and auxiliary functions of SUFB such as suctioning capacity and flexibility.

In this article, we will review the published evidence on the applications of single use and reusable bronchoscopes in bronchoscopy suites and intensive care units. Specifically, we will discuss the pros and cons of these devices as pertinent to fundamental, advanced, and therapeutic bronchoscopic interventions.

## 2. Methods

### 2.1. Study Identification

The literature search for this review was performed to evaluate the use of SUFB and RFB in the pulmonary and critical care procedures. The population, intervention, comparison, and outcome (PICO) approach was used to guide this systematic review with the following aims:Determine the advantages and disadvantages of using SUFB as compared with RFB when performing advanced diagnostic and therapeutic bronchoscopic procedures.Describe the evidence to support the use of SUFB and/or RFB when performing advanced diagnostic and therapeutic bronchoscopy.

The online database PubMed was searched for English language publications from inception through 1 July 2021. The following terms were used in PubMed advanced search engine: bronchoscopy, flexible bronchoscopy, rigid bronchoscopy, therapeutic bronchoscopy, endobronchial ultrasound, reusable flexible bronchoscope, single-use flexible bronchoscope, disposable bronchoscope, navigational bronchoscopy, bronchial thermoplasty, endobronchial valves, endobronchial lesions, brachytherapy, photodynamic therapy, cryotherapy, thermal ablation, electrocautery, laser, argon plasma coagulation, airway stent, bronchoalveolar lavage, airway management, intubation, percutaneous dilatational tracheostomy, foreign body retrieval, massive hemoptysis. Additional articles were captured after reviewing the reference lists from identified studies and pertinent review articles.

### 2.2. Study Eligibility

Articles deemed potentially eligible were divided and reviewed for eligibility according to predefined criteria.

### 2.3. Statistical Analysis

Information provided by each of the primary study authors was used to report the evaluated outcomes. No attempt was made to pool data across studies because there was substantial heterogeneity in comparator and outcome measures, populations studied, and few studies provided the individual data necessary for quantitative synthesis.

## 3. Advantages of SUFB

SUFB have several advantages over RFB. Perhaps the most noteworthy advantage is that SUFB theoretically offer complete sterility as compared with RFB. Several studies have shown transmission of pathogenic organisms via contaminated RFB despite using appropriate decontamination procedures [4,5,6,7,8]. In a multi-centered prospective study, Ofstead and colleagues reported that despite complete adherence to high-level disinfection (HLD) and reprocessing procedures, residual proteins and infectious pathogens were seen on fully reprocessed RFB that were ready for patient use. The authors concluded that HLD measures were not effective and advocated for a shift toward the use of sterilized bronchoscopes [9]. A systematic review, which included observational and retrospective data from 16 studies involving 3120 bronchoscopic procedures, reported 476 cases of cross-contamination with 86 cases of infection requiring antibiotic therapy. Six of these studies reportedly took place in endoscopy units [10,11,12,13,14,15,16]. Srinivasan and colleagues reported on a large outbreak of P. aeruginosa infections related to the use of a contaminated bronchoscope with a damaged biopsy-port cap. The group reviewed 665 bronchoscopies over an 8-month period, and reported a total of 48 respiratory tract infections, 39 bloodstream infections, and 3 deaths [16].

Several studies have suggested that SUFB are more cost-effective than RFB use in the intensive care unit and bronchoscopy suite and suggested that when taking potential costs of treating bronchoscopy-related infections into account, the use of SUFB was more cost-effective than RFB and associated with less cases of cross-contamination and bronchoscopy-associated infections requiring antibiotics [10,17]. Furthermore, since SUFB are one-time use devices, they do not require reprocessing after use, and therefore may translate to reduced cleaning and storage costs [18]. However, the clinical consequences of bronchoscopy-related cross-contamination leading to subsequent infection requiring antibiotic therapy with using SUFB instead of RFB, assuming appropriate decontamination procedures take place, are unclear from published literature. Despite the theoretical sterility that is touted of SUFB, the actual rate of clinically relevant infections related to SUFB use has not been studied, which is dependent on how the devices are handled during their use. At the time of this writing, there has not been any comparative trial evaluating the risk of clinically relevant infections due to SUFB versus RFB.

SUFBs are portable and usually easy to access. They do not require endoscopy staff to move and set up the bronchoscopy tower and scopes, allowing for immediate availability and use for unanticipated difficult airways. However, ancillary tools (syringes, biopsy forceps, endotracheal tube adaptors, bite blocks, etc.) still must be brought in as they are not included in the currently available SUFB kits on the market. Since SUFB do not require time for reprocessing, the use of SUFB for simple bronchoscopic procedures allows for the option of parallel use of RFB for advanced diagnostic and therapeutic procedures in the bronchoscopy suite, potentially avoiding delays and optimizing the timing of bronchoscopic interventions. Due to the ease of set up, SUFB confers the advantage of being used “out of hours” and outside of the bronchoscopy suite [1]. Furthermore, it is an affordable platform for off-site bronchoscopic training and research. Since SUFBs are one-time use and disposable, they reduce the potential incidence of RFB damage, thus increasing RFB availability [1,18]. The use of SUFB may offset the high cost of repairs, need for decontamination, and possible cross-contamination with subsequent bronchoscopy-related infections [19]. However, these advantages will need to be further explored with future studies.

## 4. Advantages of RFB

For several decades of advanced diagnostic and therapeutic bronchoscopic procedures, RFBs have been used given their maneuverability, handling, deflection, image quality, and adequate channel size for passing of instruments. Studies addressing advanced diagnostic bronchoscopic procedures, such as mediastinal staging, involved RFB. The same is true for many therapeutic bronchoscopic procedures such as airway stenting, cryotherapy, ablative therapy, valve placement for persistent air leak and bronchial lung volume reduction, and bronchial thermoplasty. In a survey which included attending and in-training physicians, more than half of respondents felt that RFB provided better image quality, maneuverability, suction, and medical record integration than SUFB. Additionally, 30% of respondents indicated RFB allowed for easier passage of tools and thought that RFB were cheaper than SUFB [19]. It is important to mention that despite the concerns for cross-contamination and subsequent bronchoscopy-related infections, RFB are still used for advanced diagnostic procedures such as mediastinal staging and guided bronchoscopic procedures during the pandemic. SUFB may not yet be suitable for more advanced diagnostic and therapeutic procedures until further advancements in the technology are developed.

## 5. SUFB Use for Pulmonary Procedures

Bronchoscopic procedures such as simple airway inspection and bronchoalveolar lavage can be safely performed using the SUFB (Figure 1). Zaidi and colleagues demonstrated that cell yield and viability in BAL sample were comparable between SUFB and RFB, with greater sample volumes when using SUFB [20]. The authors concluded that SUFB are an acceptable and cost-effective option when compared with standard RFB for obtaining a BAL. Of note, recent studies which evaluated the feasibility, safety, navigation success, and diagnostic yield in sampling pulmonary lesions using the Monarch and Ion robotic navigational systems, used bronchoscopes that technically speaking are single use disposable devices that are compatible with their respective platforms [21,22,23,24,25]. Although these robotic bronchoscopes are not handheld devices, the use of these single use disposable scopes suggest that there may be a role for conventional SUFB in advanced diagnostic procedures as well.

## 6. RFB Use for Pulmonary Procedures

In these authors’ experience, RFB are preferred at this time for advanced diagnostic and therapeutic procedures based on currently available technology due to improved image quality, sturdiness, handling, and suction power. These higher capabilities may be required, especially if there are intra-operative complications such as airway bleeding. Navigational bronchoscopy, endobronchial valve placement, bronchial thermoplasty, and management of airway disease with thermal ablation, cryotherapy, and airway stenting require a bronchoscope with a high degree of maneuverability and high-definition optics.

### 6.1. Endobronchial Ultrasound

There are no single use convex probe endobronchial ultrasound (CP-EBUS) bronchoscopes and given the cost of manufacturing such devices, it is unlikely these will be available in the near future. Hence, all studies involving EBUS use dedicated EBUS-TBNA reusable bronchoscopes. In addition, airway inspection before and after EBUS-TBNA, in published studies, has been performed with RFB not SUFB. For instance, a retrospective study comparing the efficacy of EBUS-TBNA with mediastinoscopy for staging in lung cancer utilized both conventional reusable flexible bronchoscopes for airway inspection and reusable EBUS bronchoscopes [26]. EBUS-TBNA studies which explored the optimal number of passes for lymph node sampling and whether suction use during EBUS-TBNA affected sample adequacy and specimen quality utilized reusable CP-EBUS bronchoscopes [27,28]. Studies which evaluated the feasibility and diagnostic yield of EBUS-TBNA for the work up of sarcoidosis and lymphoma also used reusable CP-EBUS bronchoscopes [29,30]. For the airway inspection part of these procedures, Gupta and colleagues report using conventional RFB [29], otherwise the other study protocols did not mention of using RFB or SUFB during that portion of the exam [27,28,30].

### 6.2. Pulmonary Lesion Sampling

To date, most peripheral bronchoscopy studies used RFBs. Some have evaluated the role and efficacy of standard flexible bronchoscopy with fluoroscopy and thin bronchoscopy with r-EBUS for biopsy of peripheral pulmonary lesions. One landmark multi-centered, prospective, randomized trial utilized RFB and reusable r-EBUS probes [31]. Similarly, clinical trials which investigated the safety and diagnostic yield of EMN guidance systems such as with superDimension and the SPiN System from Veran, also utilized RFB in their studies [32,33]. The large, prospective multicenter NAVIGATE trial, which evaluated diagnostic yield and adverse event rates in patients who underwent peripheral pulmonary lesion sampling with EMN guidance systems did not have protocol-specified restriction on procedural technique, but reusable CP-EBUS scopes were used and there was no mention of SUFB use in the study protocol [34,35]. Likewise, the PRECISION-1 and ALL IN ONE trials, which evaluated other navigational technologies, also used RFB [36,37]. A retrospective study which involved the evaluation of safety and diagnostic yield of intraprocedural cone-beam computed tomography (CBCT) imaging with augmented fluoroscopy, also utilized RFB as per the study protocol [38].

### 6.3. Bronchial Thermoplasty

In the multicenter, randomized, double-blind, sham-controlled clinical trial, the AIR2 Trial Study Group showed that patients with severe asthma who underwent bronchial thermoplasty had improved quality of life, with a reduction in severe exacerbations [39]. Similarly, the RISA Trial Study Group, evaluated the safety and efficacy of bronchial thermoplasty in patients with symptomatic, severe asthma [40]. Both studies involved the use of flexible bronchoscopy without indication of SUFB use in the study protocol. Increased range of angulation is often needed to reach the segmental airways of right and left upper lobes during the third session of bronchial thermoplasty. This may be achievable with SUFB, but it has not yet been reported in the published literature.

### 6.4. Endobronchial Valves

The STELVIO trial was the first study to demonstrate persistent improvement of lung function, exercise capacity and quality of life after bronchial lung volume reduction (BLVR) at 6 months and 1 year. In this study, RFB were used for endobronchial valve deployment [41]. The VENT, IMPACT, TRANSFORM, LIBERATE, and EMPROVE trials, which evaluated the safety and improvement of lung function and symptoms in patients who underwent BLVR did not delineate the use of RFB or SUFB in their study protocols [42,43,44,45,46]; but to our knowledge and in our center, only RFBs have been used in these studies. Clinical trials have also evaluated the feasibility and efficacy of EBV placement for managing persistent air leak (PAL), especially in patients who are high-risk for surgical intervention [47,48]. EBVs were deployed using flexible bronchoscopes without indication of SUFB use in the study protocols. In our institutions, we currently use conventional flexible bronchoscopes, which we believe offer maximum degree of angulation and maneuverability and quality optics which are all needed for EBV placement, especially when targeting the upper lobes.

### 6.5. Endobronchial Lesions

A multitude of bronchoscopic techniques are used for managing malignancies with endobronchial involvement. Among these, two studies which evaluated the efficacy of brachytherapy and photodynamic therapy for symptom palliation in patients with endobronchial tumor involvement, showed that both were effective tools for achieving airway patency and improving symptoms. Both studies predate the introduction of SUFB and used conventional RFBs in their studies [49,50]. Similarly, RFBs were used in studies that evaluated the utility of photodynamic therapy in patients with early stage endobronchial squamous cell carcinoma [51].

The bronchoscopic use of cryotherapy, electrocautery, Nd:YAG laser, and argon plasma coagulation (APC) have been described to manage malignant endobronchial disease. Studies evaluating their efficacy involved the use of RFBs. The large landmark retrospective study by Cavaliere and colleagues described the efficacy and safety profile of Nd:YAG via both rigid and flexible bronchoscopes in addressing endotracheal and endobronchial lesions [52]. Similarly, retrospective studies which explored the efficacy and safety profile of electrocautery, Nd:YAG, and cryotherapy involved the use of flexible bronchoscopy in their study protocols [53,54,55]. These studies predate the introduction of SUFB. Comparative trials of Nd:YAG, electrocautery and APC for management of endobronchial lesions used RFB in their protocols [56,57].

### 6.6. Airway Stenting

The effectiveness, long-term complications, and survival of patients with complex malignant airway stenosis who underwent bronchoscopic self-expandable metallic stent (SEMS) insertion were studied in several trials over the past decades. In one study protocol, both rigid and flexible bronchoscopes were used; there was no indication that SUFB were used for stent insertion [58]. A separate retrospective study evaluated the feasibility, complications, and long-term impact of using bronchoscopically deployed balloon-expandable stents for treating lobar bronchial stenosis. This study utilized RFB in its study protocol and it showed that lobar airway stenting was feasible with subsequent improved symptoms and radiographic outcomes [59]. Other clinical trials have evaluated the effectiveness of SEMS for palliative treatment of patients with malignant tracheoesophageal fistula. These studies utilized rigid bronchoscopes and RFB [60,61].

As exemplified in the above sections, studies involving advanced diagnostic and therapeutic procedures utilized RFB in their study protocols, likely because many, but not all, have been performed prior to the introduction of SUFBs. The use of SUFB was not described in any of these study protocols which involved advanced diagnostic and therapeutic procedures. This may also be due to the fact that many of these procedures require optimal image quality, sturdiness, handling, and suction power which are provided by RFB. Therefore, we suggest that until more data become available, RFB should be used for these advanced procedures. This may change as specifications for SUFB continue to improve in newer generations of such scopes.

## 7. SUFB Use in the Intensive Care Unit

Flexible bronchoscopy is often used in the intensive care unit for fiberoptic intubation, therapeutic aspiration, foreign body removal, percutaneous tracheostomy placement, and for managing massive hemoptysis. In a retrospective study of 93 patients at an ICU in a tertiary referral center in Singapore, the authors compared the use of SUFB to RFB. The study demonstrated that the mean interval time to procedure start time was shorter with SUFB (10 min) as compared with RFB (66 min), and less personnel was needed to operate and set up SUFB as compared with RFB. In this study, SUFBs were primarily used for tracheostomy placement, BAL, airway inspection, pulmonary hygiene, hemoptysis, and intubation [62]. Since SUFBs are more immediately available with little set-up time, the authors recommended using SUFBs in the intensive care unit for these procedures. Herein, we summarize the available literature on SUFBs in the ICUs.

### 7.1. Bronchoalveolar Lavage and Aspiration of Secretions

Mankikian and colleagues performed a satisfaction questionnaire after using a SUFB for bronchoalveolar lavage (BAL) in the ICU with the following results: “acceptable” to “very good” for quality of aspiration, maneuverability, and quality of vision; and “very good” to “perfect” for setting up and insertion. The authors suggested that using SUFB obviates the need for disinfection and thereby eradicates potential cross-contamination among ICU patients [63].

In a study by Gao and colleagues, bronchoscopy with BAL was performed on intubated patients with COVID-19. The SUFBs were utilized for the procedure. The study notes that at the time of the publication, more than 450 BAL samples were obtained from COVID-19 patients. The authors report that by utilizing aerosol limiting precautions and appropriate personal protective equipment (PPE), none of the 47 bronchoscopist respondents tested positive on subsequent nasopharyngeal swab testing, hence suggesting that BAL was safe to perform on COVID-19 intubated patients [64]. Chang and colleagues utilized SUFBs for 241 bronchoscopies in the ICU on mechanically ventilated COVID-19 patients. This study reported a high number of secondary infections (bacterial, fungal) with the BAL samples demonstrating a 65% positive culture rate compared with the tracheal aspirate only demonstrating a 45% culture positive rate. The authors reported safety of bronchoscopy when healthcare workers utilized appropriate PPE [65]. Diez-Ferrer and colleagues reported using SUFBs in 94 bronchoscopies performed on 51 patients in the ICU in the beginning of the pandemic and reported that thick secretions found in COVID-19 patients increased the time required for therapeutic aspiration [66]. These studies suggest that SUFBs could be useful in the ICUs for BALs and for secretion management, including patients suffering from COVID-19-related respiratory failure. Of note, at the time of this writing, there are no studies or published guidelines for reusing SUFB within a short period of time for the same patient (e.g., for aspiration of secretions in the ICU).

### 7.2. Airway Management

Initial studies which involved earlier versions of SUFB reported lower success rates of fiberoptic intubation as compared with using RFB [67]. This was primarily due to low image resolution, poor maneuverability, and the lack of suction capability. Since then, newer generations of SUFB with improved optics, handling, and suction capability have been developed. Various studies have shown that fiberoptic intubations with SUFB are acceptable and comparable with RFB in the anesthesia setting [68,69,70,71,72,73]. In a study mimicking a difficult airway by immobilizing patients with semi-rigid collars, the authors compared fiberoptic intubation with SUFB and RFB in 100 subjects. The authors demonstrated successful intubation with either bronchoscope in all cases but noted that the median time with SUFB intubation was longer than with RFB. However, the authors pointed out that it was much quicker to transport and set up SUFB than RFB, which may offset the time to intubation [74]. A prospective, multicentered non-interventional study evaluated operator preference regarding SUFB use in 176 patients undergoing intubation or bronchoscopy in the intensive care setting. In this analysis, the authors found that there was an overall preference for SUFB over RFB for both intubation and bronchoscopy [75].

### 7.3. Percutaneous Tracheostomy Guidance

Reynolds and colleagues described successful percutaneous tracheostomy placement using SUFB in a cohort of 22 patients. In one procedure, the authors had to convert to RFB due to airway bleeding [76]. The rate of subsequent procedural complications associated with percutaneous tracheostomy placement such as tracheal stenosis and fractured cartilaginous rings was not reported. Niroula and colleagues performed a retrospective review of patients that received a percutaneous tracheostomy during the COVID-19 pandemic in a single center using a modified protocol of apnea in 28 cases. SUFBs were successfully utilized in 19 of those cases. The authors mention that a RFB was utilized in the remaining cases due to lack of availability of the SUFB [77]. In our institution we use both SUFB and RFB for guidance during percutaneous tracheostomy. The choice of scope is operator dependent and often based on scope and staff availability (Figure 2).

## 8. RFBs in the Intensive Care Unit

### 8.1. Foreign Body Removal

Flexible bronchoscopy has been shown to be a useful tool for successful airway foreign body extraction. The available literature describes the use of RFB for this purpose [78,79,80]. A small case series demonstrated safety and successful use of cryoextraction of foreign bodies. This case series utilized RFB for foreign body retrieval [81]. While not described in the literature, we suggest that airway foreign bodies can also be extracted utilizing SUFB, as long as the working channel allows the insertion of necessary tools (grasping forceps, baskets) and scope maneuverability and imaging allows operators to safely perform the desired maneuvers.

### 8.2. Management of Massive Hemoptysis

Massive hemoptysis in the ICU generally focuses on airway management to ensure ongoing adequate oxygenation, ventilation, bleeding localization, and avoidance of spillage into the contralateral lung by using bronchial blockers or selective intubation techniques [82] (Figure 3). Bronchoscopic hemostasis techniques include the use of vasoconstrictive approaches such as cold saline, or using argon plasma coagulation (APC), laser, or electrocautery. Other techniques for the management of hemoptysis include bronchial artery embolization, in which case bronchoscopy can help by localizing the source and thus potentially improving the success of BAE [83]. While emergent airway management techniques have been well described utilizing SUFBs as mentioned above, to date, there is no literature describing the use of SUFBs for the purpose of APC, laser, or electrocautery. We believe that these procedures require scopes with good suction, easy handling, and optimal imaging given the acuity of the problem and suboptimal field of view in the setting of airway hemorrhage.

### 8.3. Percutaneous Tracheostomy Guidance

Bronchoscopic guidance for the placement of percutaneous tracheostomy at the bedside in the ICU has been well described. While initial description of percutaneous tracheostomy placement by Ciaglia and colleagues in 1985 was a “blind” technique, the endoscopic approach for visualization of the procedure was subsequently described in 1989 [84,85]. Most studies used RFBs for the placement of percutaneous tracheostomies. Recently, investigators have reported the use of SUFBs for the use of percutaneous tracheostomies [75] and to our knowledge, this is now commonly done during the pandemic. However, in a large study involving percutaneous tracheostomy placement in 98 patients with SARS-CoV-2, Angel and colleagues describe a novel percutaneous tracheostomy technique to minimize aerosolization. In the described technique, the bronchoscope is advanced adjacent to the endotracheal tube instead of the standard practice of advancing the bronchoscope through the endotracheal tube. The authors mention that a therapeutic RFB was used to ensure that enough torque and rigidity was available to maneuver the scope around the endotracheal tube and push the scope through the anterior space between the vocal cords and the ETT [86]. A cost comparison analysis was performed with using SUFB versus RFB for the placement of percutaneous tracheostomy. In this study, the authors concluded that significant savings can be made using the SUFBs to guide percutaneous tracheostomy placement due to the costs of reprocessing along with repair costs [87].

## 9. Approach to Flexible Bronchoscopy in the Intensive Care Unit

Acquiring the practical skills, knowledge, and experience needed to perform flexible bronchoscopy proficiently in the critical care setting can be a challenging task. Solidoro and colleagues provide a framework for obtaining the skills, knowledge, and experience needed for flexible bronchoscopy in the ICU setting. The authors propose training which incorporates both theoretical knowledge and obtaining practical skills of flexible bronchoscopy in the ICU setting via training on animal or cadaver models as well as utilizing both low- and high-fidelity simulators. The authors suggest the use of both qualitative and quantitative assessments to measure competency by measuring the ability to perform specific tasks using flexible bronchoscopy in the ICU along with maintaining a procedure log [88].

## 10. Comparative Trials of SUFBs and RFBs

The SUFBs will very likely be more commonly used in the bronchoscopy suites, operating rooms, and ICUs. This is due to the reportedly lower costs and lower cross-contamination or infection rates of 0% versus 2.8% reported for SUFBs versus RFBs, respectively [10]. There are at least three SUFBs available in the market [18], but only a few studies compared these devices among themselves or with the RFBs. Most studies are satisfaction surveys or bench model testing. For example, a user satisfaction study in Spain showed that one of the newest SUFB models received high ratings for ease of use, imaging, and suction [89]. In a simulation study, a slim model of a SUFB required more time for nasal intubation than a RFB. This was thought to be due to the need for more scope rigidity for the management of difficult airways [73]. A comparative analysis from 2018 concluded that SUFBs result in decreased patient waiting time, are better for teaching, and offer increased safety for hospital staff, but could become more costly compared to RFBs in high volume practices [90]. We and others suggest that cost analyses must consider the incidence of breakdown of RFBs and the cost of disinfection procedures, and the cost associated with treating clinically relevant bronchoscopy-related infections. Future studies will have to systematically evaluate new SUFBs with the currently available RFBs and among themselves regarding clinically meaningful outcomes and costs.

## 11. Conclusions

The portability, immediate availability, and theoretical reduced risk of clinically relevant infections confer an advantage of using SUFB over RFB in certain scenarios in the bronchoscopy and intensive care units. However, despite the reported economic advantages and decreased risk for infection transmission when using SUFB, studies have not yet been performed for more complex bronchoscopic procedures. To date, there are no comparative studies that have demonstrated equivalent capability with respect to flexibility, angulation, image quality between SUFB and RFB. In addition, to date, RFB have been used for the management of massive hemoptysis, tracheal stenosis, endobronchial obstruction, staging and diagnosis of peripheral lung lesions. Therefore, based on the data available as of this writing, we conclude that the use of SUFB should be limited to flexible bronchoscopic intubation, simple therapeutic aspirations, BAL, and in low-risk percutaneous tracheostomy procedures until further evidence for more widespread use becomes available (Table 1). For SUFB to gain wider adoption and extend into more complex interventional procedures, there will be a need for significant investment from manufacturers to improve on the technology or to develop different scopes for different purposes. Ongoing improvements in maneuverability, larger inner channel size, angle tip deflection, sturdiness, and image quality of these devices to match more closely with the specifications and features of RFB will be critical for a broader adoption of these devices [91]. The reliability and clinical impact of using newer generation SUFB in more complex procedures remains to be determined.

## Figures and Tables

**Figure 1 diagnostics-12-00174-f001:**
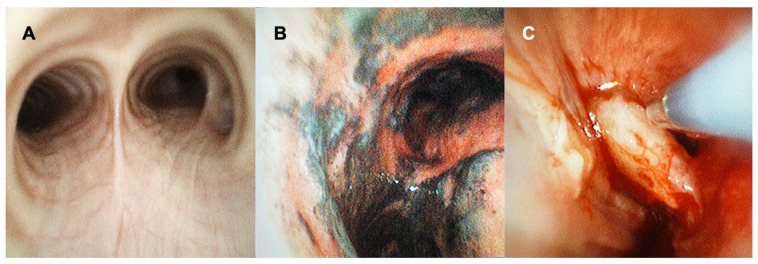
Using SUFB for Airway Inspection and Cryotherapy. This figure demonstrates the use of SUFB (EXALT Model B Single-Use Bronchoscope from Boston Scientific). (**A**) Airway inspection prior to a robotic bronchoscopy procedure for sampling peripheral lung lesions. Secretions were easily suctioned until clean in airways using the SUFB. (**B**) Airway inspection with the view of the main carina using the SUFB in a patient with airway burn. (**C**) Visualization of the airway during cryotherapy with the SUFB.

**Figure 2 diagnostics-12-00174-f002:**
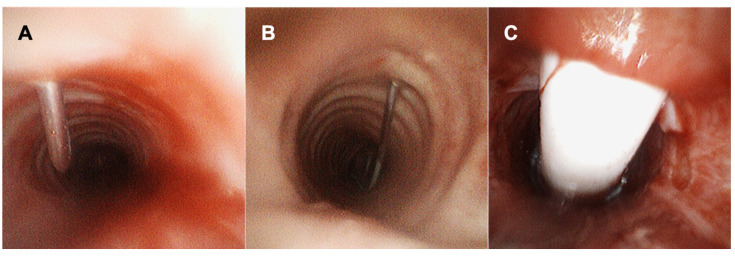
Using SUFB for Percutaneous Tracheostomy Tube Placement. This figure illustrates tracheostomy placement using the SUFB (EXALT Model B Single-Use Bronchoscope from Boston Scientific). Needle insertion (**A**), passing of the guidewire (**B**), and placement of the tracheostomy tube (**C**) are visualized.

**Figure 3 diagnostics-12-00174-f003:**
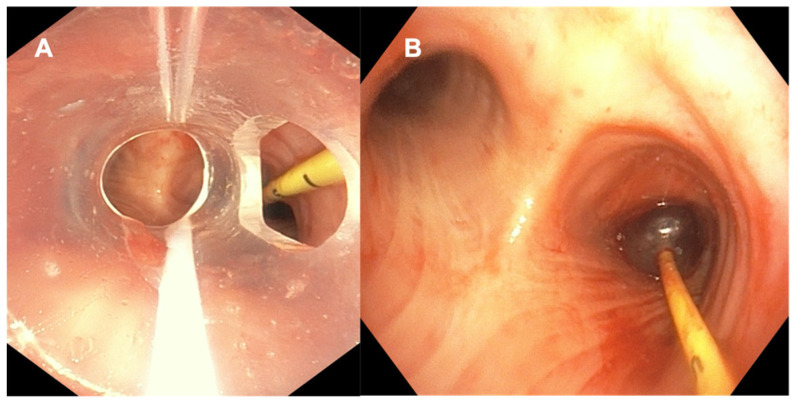
Using RFB for Management Hemoptysis. These figures illustrate high-quality images of the endobronchial blocker next to the endotracheal tube (**A**) and isolating the bronchus intermedius (**B**) using RFB (Olympus P 190 bronchoscope).

**Table 1 diagnostics-12-00174-t001:** Recommended indications for SUFB and RFB.

	Diagnostic	Therapeutic
SUFB	BAL Simple airway inspection Washings	Flexible bronchoscopic intubation Therapeutic aspiration Percutaneous tracheostomy *
RFB	Peripheral nodule sampling Mediastinal sampling Complex airway inspection	Thermal ablation Cryotherapy Debulking Bronchial thermoplasty Endobronchial valve placement Massive hemoptysis Foreign body retrieval Airway stent placement Fiducial marker placement Percutaneous tracheostomy *

* Authors recommend that SUFB use should be reserved for low-risk percutaneous tracheostomy placement procedures until further evidence is available.

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
