# Peer review of "Single-Use and Reusable Flexible Bronchoscopes in Pulmonary and Critical Care Medicine"

_diagnostics, 2022, doi:10.3390/diagnostics12010174_

Round 1

Reviewer 1 Report

Dear Authors

thank you for your valuable submission.

As a general consideration, i believe it could be useful to add a short methodologic paragraph, indicating the criteria you adopted for literature search and retrieval. This may include time lag, used databases and criteria for inclusion/exclusion, and it would add "reliability" to your already rich and informative review.

Introduction, line 44: please expand the acronyms of different societies.

paragraph for use in ICU: i believe on of the most important issues for non-pulmonologist and in general with physicians approaching the FB is experience and skill acquisition, and this concept deserves, in my opinion, to be highlighted. You may also reference with this paper: Solidoro P, Corbetta L, Patrucco F, et Al. Competences in bronchoscopy for Intensive Care Unit, anesthesiology, thoracic surgery and lung transplantation. Panminerva Med. 2019 Sep;61(3):367-385. doi: 10.23736/S0031-0808.18.03565-6.

In table I, I would change "fiberoptic intubation" for "flexible optic intubation" or similar, given that newer devices, either disposable or reusable, not always use optical fibers anymore.

References list is updated and generous, once again i would recommend a paragraph describing the search and inclusion criteria.

Author Response

Reviewer #1:

  1. As a general consideration, I believe it could be useful to add a short methodologic paragraph, indicating the criteria you adopted for literature search and retrieval. This may include time lag, used databases and criteria for inclusion/exclusion, and it would add "reliability" to your already rich and informative review.
    1. A “Methods” section has been added. This is reflected in lines 61-93.
  2. Introduction, line 44: please expand the acronyms of different societies.
    1. The acronyms are now expanded to reflect the names of the societies. This is addressed in lines 44, 45, 50.
  3. Paragraph for use in ICU: I believe one of the most important issues for non-pulmonologist and in general with physicians approaching the FB is experience and skill acquisition, and this concept deserves, in my opinion, to be highlighted.
    1. A separate section “Approach to Flexible Bronchoscopy in the Intensive Care Unit” has been added. This is reflected in lines 419-429.
    2. References have been added and adjusted accordingly. New reference has been added in lines 735-737.
  4. In table I, I would change "fiberoptic intubation" for "flexible optic intubation" or similar, given that newer devices, either disposable or reusable, not always use optical fibers anymore.
    1. This is now addressed. “Fiberoptic intubation” is changed to “Flexible bronchoscopic intubation” in Table 1 to more accurately reflect the procedure and equipment used.

Reviewer 2 Report

This manuscript reviews single-use flexible bronchoscopes (SUFB) in respiratory and critical care medicine. Although it has the minor drawback that few clinical trials have investigated the usefulness of SUFB, the manuscript provides a concise and easy-to-understand summary of the current situation. Please consider additional statements for the following items.

  1. Some extra hyphenation in abstracts, mainly due to editorial issues (at-ten-tion, pub-lished, interven-tions)
  2. How should we consider the re-use of the device in a short period of time for the same patient (e.g., for aspiration of sputum in the ICU); how many times and for how long can it be used?

Author Response

Reviewer #2:

  1. Some extra hyphenation in abstracts, mainly due to editorial issues (at-ten-tion, pub-lished, interven-tions)
    1. This is now addressed. The extra hyphenation in the abstract due to editorial issues have been fixed, and reflected in lines 14, 23, and 26.
  2. How should we consider the re-use of the device in a short period of time for the same patient (e.g., for aspiration of sputum in the ICU); how many times and for how long can it be used?
    1. At the time of this writing, there is currently no studies or published guidelines on the reuse of SUFB for the same patient. This is reflected in lines 329-331.